# Δ-encoder: an effective sample synthesis method for few-shot object recognition

Eli Schwartz*[1,2], Leonid Karlinsky*[1],
Joseph Shtok[1], Sivan Harary[1], Mattias Marder[1], Abhishek Kumar[1],
Rogerio Feris[1], Raja Giryes[2] and Alex M. Bronstein[3]

[1]IBM Research AI
[2]School of Electrical Engineering, Tel-Aviv University, Tel-Aviv, Israel
[3]Department of Computer Science, Technion, Haifa, Israel

## Abstract

Learning to classify new categories based on just one or a few examples is a long-standing challenge in modern computer vision. In this work, we propose a simple yet effective method for few-shot (and one-shot) object recognition. Our approach is based on a modified auto-encoder, denoted Δ-encoder, that learns to synthesize new samples for an unseen category just by seeing few examples from it. The synthesized samples are then used to train a classifier. The proposed approach learns to both extract transferable intra-class deformations, or "deltas", between same-class pairs of training examples, and to apply those deltas to the few provided examples of a novel class (unseen during training) in order to efficiently synthesize samples from that new class. The proposed method improves the state-of-the-art of one-shot object-recognition and performs comparably in the few-shot case.

Corresponding author: Leonid Karlinsky (`leonidka@il.ibm.com`)

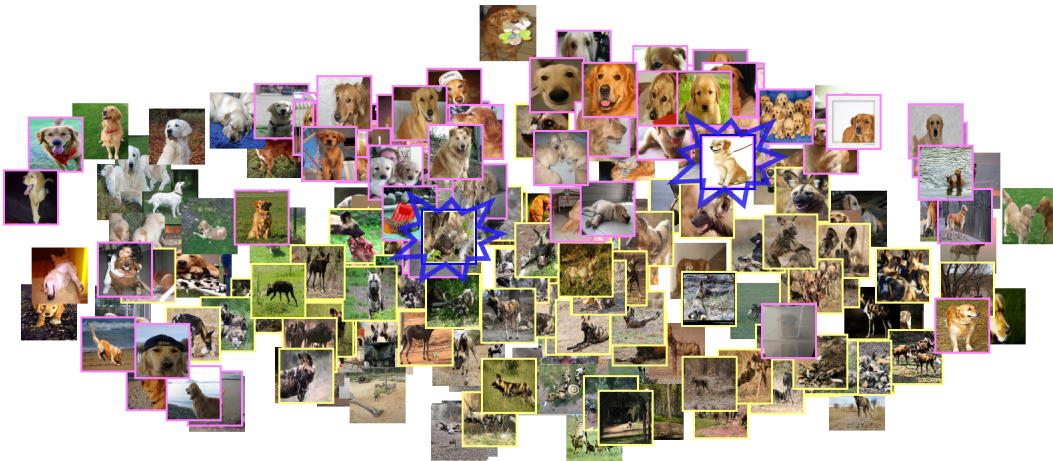

Figure 1: **Visualization of two-way one-shot classification** trained on synthesized examples. *Correctly* classified images are framed in magenta (Golden retriever) and yellow (African wild dog). The only two images seen at training time and used for sample synthesis are framed in blue. Note the non-trivial relative arrangement of examples belonging to different classes handled successfully by our approach. The figure is plotted using t-SNE applied to VGG features. Best viewed in color.

# 1 Introduction

Following the great success of deep learning, the field of visual classification has made a significant leap forward, reaching – and in some cases, surpassing – human levels performance (usually when expertise is required) [24, 37]. Starting from AlexNet [23], followed by VGG [38], Google Inception [42], ResNet [18], DenseNet [20] and NASNet [54] the field made tremendous advances in classification performance on large-scale datasets, such as ImageNet [5], with thousands of examples per category. However, it is known that we humans are particularly good at learning new categories on the go, from seeing just a few or even a single example [24]. This is especially evident in early childhood, when a parent points and names an object and a child can immediately start finding more of its kind in the surroundings.

While the exact workings of the human brain are very far from being fully understood, one can conjecture that humans are likely to learn from analogies. That is, we identify in new objects elements of some latent semantic structure, present in other, already familiar categories, and use this structure to construct our internal classifier for the new category. Similarly, in the domain of computer vision, we assume that we can use the plentiful set of examples (instances) of the known classes (represented in some latent semantic space), in order to *learn to sample* from the distributions of the new classes, the ones for which we are given just one or a few examples.

Teaching a neural network to sample from distributions of new visual categories, based on just a few observed examples, is the essence of our proposed approach. First, the proposed approach learns to extract and later to sample (synthesize) transferable non-linear deformations between pairs of examples of seen (training) classes. We refer to these deformations as "deltas" in the feature space. Second, it learns to apply those deltas to the few provided examples of novel categories, unseen during training, in order to efficiently synthesize new samples from these categories. Thus, in the few-shot scenario, we are able to synthesize enough samples of each new category to train a classifier in the standard supervised fashion.

Our proposed solution is a simple, yet effective method (in the light of the obtained empirical results) for learning to sample from the class distribution after being provided with one or a few examples of that class. It exhibits improved performance compared to the state-of-the-art methods for few-shot classification on a variety of standard few-shot classification benchmarks.

# 2 Related work

**Few-shot learning by metric learning:** a number of approaches [47, 39, 36] use a large corpus of instances of known categories to learn an embedding into a metric space where some simple (usually $L_2$) metric is then used to classify instances of new categories via proximity to the few labeled training examples embedded in the same space. In [13], a metric learning method based on graph neural networks, that goes beyond the $L_2$ metric, have been proposed. The metric-learning-based approaches are either posed as a general discriminative distance metric learning (DML) scheme [36], or optimized to operate in the few shot scenario [39, 47, 13]. These approaches show great promise, and in some cases are able to learn embedding spaces with quite meaningful semantics embedded in the metric [36]. Yet, their performance is in many cases inferior to the meta-learning and generative (synthesis) approaches that will be discussed next.

**Few-shot meta-learning (learning-to-learn):** these approaches are trained on few-shot tasks instead of specific object instances, resulting in models that once trained can "learn" on new such tasks with relatively few examples. In Matching Networks [43], a non-parametric $k$-NN classifier is meta-learned such that for each few-shot task the learned model generates an adaptive embedding space for which the task can be better solved. In [39] the embedding space is optimized to best support task-adaptive category population centers (assuming uni-modal category distributions). In approaches such as MAML [10], Meta-SGD [26], DEML+Meta-SGD [52], Meta-Learn LSTM [34] and Meta-Networks [31], the meta-learned classifiers are optimized to be easily fine-tuned on new few-shot tasks using small training data.

**Generative and augmentation-based few-shot approaches:** In this line of methods, either generative models are trained to synthesize new data based on few examples, or additional examples are obtained by some other form of transfer learning from external data. These approaches can be categorized as follows: (1) semi-supervised approaches using additional unlabeled data [6, 11]; (2)

fine tuning from pre-trained models [25, 45, 46]; (3) applying domain transfer by borrowing examples from relevant categories [27] or using semantic vocabularies [2, 12]; (4) rendering synthetic examples [32, 7, 40]; (5) augmenting the training examples [23]; (6) example synthesis using Generative Adversarial Networks (GANs) [53, 21, 14, 35, 33, 29, 8, 20]; and (6) learning to use additional semantic information (e.g. attribute vector) per-instance for example synthesis [4, 51]. It is noteworthy that all the augmentation and synthesis approaches can be used in combination with the metric learning or meta-learning schemes, as we can always synthesize more data before using those approaches and thus (hopefully) improve their performance.

Several insightful papers have recently emerged dealing with sample synthesis. In [17] it is conjectured that the relative linear offset in feature space between a pair of same-class examples conveys information on a valid deformation, and can be applied to instances of other classes. In their approach, similar (in terms of this offset) pairs of examples from different categories are mined during training and then used to train a generator optimized for applying the same offset to other examples. In our technique, we do not restrict our "deltas" to be linear offsets, and in principle can have the encoder and the generator to learn more complex deformations than offsets in the feature space.

In [44], a generator sub-network is added to a classification network in order to synthesize additional examples on the fly in a way that helps training the classifier on small data. This generator receives the provided training examples accompanied by noise vectors (source of randomness). At the learning stage, the generator is optimized to perform random augmentation, jointly with the meta-learner parameters, via the classification loss. In contrast, in our strategy the generator is explicitly trained, via the reconstruction loss, to transfer deformations between examples and categories. A similar idea of learning to randomly augment class examples in a way that will improve classification performance is explored in [1] using GANs. In [35], a few-shot class density estimation is performed with an autoregressive model, augmented with an attention mechanism, where examples are synthesized by a sequential process. Finally, the idea of learning to apply deformations on class examples has also been successfully explored in other domains, such as text synthesis [16].

## 3    The $\Delta$-encoder

We propose a method for few-shot classification by learning to synthesize samples of novel categories (unseen during training) when only a single or a few real examples are available. The generated samples are then used to train a classifier. Our proposed approach, dubbed as the $\Delta$-encoder, learns to sample from the category distribution, while being seeded by only one or few examples from that distribution. Doing so, it belongs to the family of example synthesis methods. Yet, it does not assume the existence of additional unlabeled data, e.g., transferable pre-trained models (on an external dataset) or any directly related examples from other categories or domains, and it does not rely on additional semantic information per-instance.

The proposed solution is to train a network comprised of an encoder and a decoder. The encoder learns to extract transferable deformations between pairs of examples of the same class, while the decoder learns how to apply these deformations to other examples in order to learn to sample from new categories. For the ease of notation, assume we are given a single example $Y$ belonging to a certain category $\mathcal{C}$, and our goal is to learn to sample additional examples $X$ belonging to the same category. In other words, we would like to learn to sample from the class posterior: $\mathbb{P}(X|\mathcal{C}, Y)$. Notice that the conditioning on $Y$ implies that we may not learn to sample from the whole class posterior, but rather from its certain subset of "modes" that can be obtained from $Y$ using the deformations we learned to extract. Our method is inspired by the one used for zero-shot classification in [3], where the decoder is provided side information about the class, in the form of human-annotated attributes.

Our generative model is a variant of an Auto-Encoder (AE). Standard AE learns to reconstruct a signal $X$ by minimizing $\|X - \hat{X}\|_1$, where $\hat{X} = D(E(X))$ is the signal reconstructed by the AE, and $E$ and $D$ are the encoder and decoder sub-networks, respectively. A common assumption for an AE is that the intermediate bottleneck representation $E(X)$, can be of much lower dimension than $X$. This is driven by assuming the ability to extract the "semantic essence" of $X$ – a minimal set of identifying features of $X$ necessary for the reconstruction. The simple key idea of this work is to change the meaning of $E(X)$ from representing the "essence" of $X$, to representing the delta, or "additional information" needed to reconstruct $X$ from $Y$ (an observed example from the same category). To this end, we propose the training architecture depicted in Figure 2a. The encoder gets

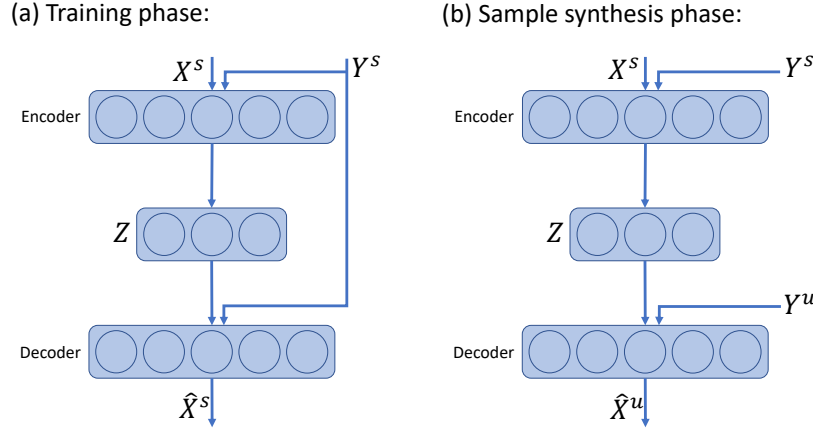

(a) Training phase:  (b) Sample synthesis phase:

Figure 2: **Proposed $\Delta$-encoder architecture.** (a) Training phase: $X^s$ and $Y^s$ are a random pair of samples from the same seen class; the $\Delta$-encoder learns to reconstruct $X^s$. (b) Sample synthesis phase: $X^s$ and $Y^s$ are a random pair of samples from a random seen class, and $Y^u$ is a single example from a novel unseen class; the $\Delta$-encoder generates a new sample $\hat{X}^u$ from the new class.

as an input both the signal $X$ and the "anchor" example $Y$ and learns to compute the representation of the additional information $Z = E(X, Y)$ needed by the decoder $D$ in order to reconstruct the $X$ from both $Y$ and $Z$. Keeping the dimension of $Z$ small, we ensure that the decoder $D$ cannot use just $Z$ in order to reconstruct $X$. This way, we regularize the encoder to strongly rely on the anchor example $Y$ for the reconstruction, thus, enabling synthesis as described next.

Following training, at the sample synthesis phase, we use the trained network to sample from $\mathbb{P}(X|\mathcal{C}, Y)$. We use the non-parametric distribution of $Z$ by sampling random pairs $\{X^s, Y^s\}$ from the classes seen during training (such that $X^s$ and $Y^s$ belong to the same category) and generating from them $Z = E(X^s, Y^s)$ using the trained encoder. Thus, we end up with a set of samples $\{Z_i\}$. In each of the one-shot experiments, for a novel unseen class $\mathcal{U}$ we are provided with an example $Y^u$, from which we synthesize a set of samples for the class $\mathcal{U}$ using our trained generator model: $\{D(Z_i, Y^u)\}$. The process is illustrated in Figure 2b. Finally, we use the synthesized samples to train a linear classifier (one dense layer followed by softmax). As a straightforward extension, for $k$-shot learning we repeat the process $k$ times, independently synthesizing samples based on each of the $k$ examples provided.

## 3.1 Implementation details

In all the experiments, images are represented by pre-computed feature vectors. In all our experiments we are using the VGG16 [38] or ResNet18 [18] models for feature extraction. For both models the head, i.e., the layers after the last convolution, is replaced by two fully-connected layers with 2048 units with ReLU activations. The features used are the 2048-dimensional outputs of the last fully-connected layer. Following the ideas of [28], we augment the $\mathcal{L}_1$ reconstruction loss ($\|X - \hat{X}\|_1$) to include adaptive weights: $\sum_i w_i |X_i - \hat{X}_i|$, where $w_i = |X_i - \hat{X}_i|^2 / \|X - \hat{X}\|_2$, encouraging larger gradients for feature dimensions with higher residual error. The encoder and decoder sub-networks are implemented as multi-layer perceptrons with a single hidden layer of 8192 units, where each layer is followed by a leaky ReLU activation ($\max(x, 0.2 \cdot x)$). The encoder output $Z$ is 16-dimensional. All models are trained with Adam optimizer with the learning rate set to $10^{-5}$. Dropout with 50% rate is applied to all layers. In all experiments 1024 samples are synthesized for each unseen class. The $\Delta$-encoder training takes about 10 epochs to reach convergence; each epoch takes about 20 seconds running on an Nvidia Tesla K40m GPU (48K training samples, batch size 128). The data generation phase takes around 0.1 seconds per 1024 samples. The code is available here.

# 4   Results

We have evaluated the few-shot classification performance of the proposed method on multiple datasets, which are the benchmarks of choice for the majority of few-shot learning literature, namely: *mini*ImageNet, CIFAR-100, Caltech-256, CUB, APY, SUN and AWA2. These datasets are common benchmarks for the zero- and few-shot object recognition, and span a large variety of properties including high- and low-resolution images, tens to hundreds of fine- and coarse-grained categories, etc. We followed the standard splits used for few-shot learning for the first four datasets; for the other datasets that are not commonly used for few-shot, we used the split suggested in [49] for zero-shot learning. Table 3 summarizes the properties of the tested datasets.

In all of our experiments, the data samples $X$ and $Y$ are feature vectors computed by a pre-trained neural-network. The experimental protocol in terms of splitting of the dataset into disjoint sets of training and testing classes is the same as in all the other works evaluated on the same datasets. We use the VGG16 [38] backbone network for computing the features in all of our experiments except those on Caltech-256 and CUB. For these small-scale datasets we used ResNet18 [18], same as [4], to avoid over-fitting. We show that even in this simple setup of using pre-computed feature vectors, competitive results can be obtained by the proposed method compared to the few-shot state-of-the-art. Combining the proposed approach with an end-to-end training of the backbone network is an interesting future research direction beyond the scope of this work.

As in compared approaches, we evaluate our approach by constructing "few-shot test episode" tasks. In each test episode for the $N$-way $k$-shot classification task, we draw $N$ random unseen categories, and draw $k$ random samples from each category. Then, in order to evaluate performance on the episode, we use our trained network to synthesize a total of 1024 samples per category based on those $k$ examples. This is followed by training a simple linear $N$-class classifier over those $1024 \cdot N$ samples, and finally, the calculation of the few-shot classification accuracy on a set of $M$ real (query) samples from the tested $N$ categories. In our experiments, instead of using a fixed (large) value for $M$, we simply test the classification accuracy on all of the remaining samples of the $N$ categories that were not used for one- or few-shot training. Average performance on 10 such experiments is reported.

## 4.1   Standard benchmarks

For *mini*ImageNet, CIFAR100, CUB and Caltech-256 datasets, we evaluate our approach using a backbone network (for computing the feature vectors) trained from scratch on a subset of categories of each dataset. For few-shot testing, we use the remaining unseen categories. The proposed synthesis network is trained on the same set of categories as the backbone network. The experimental protocol used here is the same as in all compared methods.

The performance achieved by our approach is summarized in Table 1; it competes favorably to the state-of-the-art of few-shot classification on these datasets. The performance of competing methods is taken from [4]. We remark in the table whenever a method uses some form of additional external data, be it training on an external large-scale dataset, using word embedding applied to the category name, or using human-annotated class attributes.

## 4.2   Additional experiments using a shared pre-trained feature extracting model

For fair comparison, in the experiments described above in Section 4.1 we only trained our feature extractor backbone on the subset of training categories of the target dataset (same as in other works). However, it is nonetheless interesting to see how our proposed method performs in a realistic setting of having a single pre-trained feature extractor backbone trained on a large set of external data. To this end, we have conducted experiments on four public datasets (APY, AWA2, CUB and SUN), where we used features obtained from a VGG16 backbone pre-trained on ImageNet. The unseen test categories were verified to be disjoint from the ImageNet categories in [49] that dealt with dataset bias in zero-shot experiments. The results of our experiments as well as comparisons to some baselines are summarized in Table 2. The experiments in this section illustrate that the proposed method can strongly benefit from better features trained on more data. For CUB with "stronger" ImageNet features (last column in Table 2) we achieved more than $10\%$ improvement over training only using a subset of CUB categories (last column in Table 1).

Table 1: 1-shot/5-shot 5-way accuracy results

| Method | *mini*ImageNet | CIFAR100 | Caltech-256 | CUB |
|---|---|---|---|---|
| Nearest neighbor (baseline) | 44.1 / 55.1 | 56.1 / 68.3 | 51.3 / 67.5 | 52.4 / 66.0 |
| MACO [19] | 41.1 / 58.3 | - | - | 60.8 / 75.0 |
| Meta-Learner LSTM [34] | 43.4 / 60.6 | - | - | 40.4 / 49.7 |
| Matching Nets [43] | 46.6 / 60.0 | 50.5 / 60.3 | 48.1 / 57.5 | 49.3 / 59.3 |
| MAML [10] | 48.7 / 63.1 | 49.3 / 58.3 | 45.6 / 54.6 | 38.4 / 59.1 |
| Prototypical Networks [39] | 49.4 / 68.2 | - | - | - |
| SRPN [30] | 55.2 / 69.6 | - | - | - |
| RELATION NET [41] | 57.0 / 71.1 | - | - | - |
| DEML+Meta-SGD$^\heartsuit$ [52] | 58.5 / 71.3 $^\diamond$ | 61.6 / 77.9 $^\diamond$ | 62.2 / 79.5 $^\diamond$ | 66.9 / 77.1 $^\diamond$ |
| Dual TriNet$^\heartsuit$ [4] | 58.1 / **76.9** $^\dagger$ | 63.4 / 78.4 $^\dagger$ | 63.8 / 80.5 $^\dagger$ | 69.6 / **84.1** $^\star$ |
| $\Delta$-encoder$^\heartsuit$ | **59.9** / 69.7 | **66.7** / **79.8** | **73.2** / **83.6** | **69.8** / 82.6 |

$\diamond$ Also trained on an a large external dataset    $\dagger$ Using label embedding trained on large corpus
$\star$ Using human annotated class attributes    $\heartsuit$ Using ResNet features

Table 2: 1-shot/5-shot 5-way accuracy with ImageNet model features (trained on disjoint categories)

| Method | AWA2 | APY | SUN | CUB |
|---|---|---|---|---|
| Nearest neighbor (baseline) | 65.9 / 84.2 | 57.9 / 76.4 | 72.7 / 86.7 | 58.7 / 80.2 |
| Prototypical Networks | 80.8 / 95.3 | 69.8 / 90.1 | 74.7 / **94.8** | 71.9 / 92.4 |
| $\Delta$-encoder | **90.5 / 96.4** | **82.5 / 93.4** | **82.0** / 93.0 | **82.2 / 92.6** |

## 4.3 Ablation study: evaluating different design choices

In this section we review and evaluate different design choices used in the architecture and the approach proposed in this paper. For this ablation study we use the performance estimates for the AWA, APY, SUN and CUB datasets to compare the different choices.

In [3] the authors suggested the usage of Denoising-Autoencoder (DAE) for zero-shot learning (Fig. 3a). The noise is implemented as 20% dropout on the input. In training time, the DAE learns to reconstruct $X^s$ from it's noisy version, where the decoder uses the class attributes to perform the reconstruction. At test time, the decoder is used to synthesize examples from a novel class using its attributes vector and a random noise vector $Z$. Average accuracy for the zero-shot task is $64.4\%$ (first row in Table 4). As a first step towards one-shot learning, we have tested the same architecture but using another sample from the same class instead of the attributes vector (Figure 3b). The intuition behind this is that the decoder will learn to reconstruct the class instances by editing another instances from the same class instead of relying on the class attributes. This already yields a significant improvement to the average accuracy bringing it to $81.1\%$, hinting that even a single class instance conveys more information than the human chosen attributes for the class in these datasets.

Next, we replaced the random sampling of $Z$ with a non-parametric density estimate of it obtained from the training set. Instead of sampling entries of $Z \sim N(0, 1)$, we randomly sample an instance $X^s$ belonging to a randomly chosen training class and run it through the encoder to produce $Z = E(X^s)$. This variant assumes that the distribution of $Z$ is similar between the seen and unseen classes. We observed a slight improvement of $0.5\%$ due to this change. We also tested a variant where no noise is injected to the input, i.e. replacing Denoising-Autoencoder with Autoencoder. Since we did not observe a change in performance we chose the Autoencoder for being the simpler of the two. Finally, to get to our final architecture as described in Section 3 we add $Y$ as input to the encoder. This improved the performance by $2.4\%$.

**Linear offset delta** To evaluate the effect of the learned non-linear $\Delta$-encoder we also experimented with replacing it with a linear "delta" in the embedding space. In this experiment we set $Z = E(X^s, Y^s) = X^s - Y^s$ and $\hat{X} = D(Z, Y^u) = Y^u + Z$. That means we sample linear shifts from same-class pairs in the training set and use them to augment the single example of a new class $Y^u$ that we have. For this experiment we got $\sim 10$ points lower accuracy compared to $\Delta$-encoder, showing the importance of the learned non-linear "delta" encoding.

Table 3: Summary of the datasets used in our experiments

| Dataset | Fine grained | Image size | Total # images | Seen classes | Unseen classes |
|---|---|---|---|---|---|
| *mini*ImageNet [43] | ✗ | Medium | $60K$ | 80 | 20 |
| CIFAR-100 [22] | ✗ | Small | $60K$ | 80 | 20 |
| Caltech-256 Object Category [15] | ✗ | Large | $30K$ | 156 | 50 |
| Caltech-UCSD Birds 200 (CUB) [48] | ✓ | Large | $12K$ | 150 | 50 |
| Attribute Pascal & Yahoo (aPY) [9] | ✗ | Large | $14K$ | 20 | 12 |
| Scene UNderstanding (SUN) [50] | ✓ | Large | $14K$ | 645 | 72 |
| Animals with Attributes 2 (AWA2) [49] | ✗ | Large | $37K$ | 40 | 10 |

Table 4: Evaluating different design choices. All the numbers are one-shot accuracy in %.

| Method | AWA2 | APY | SUN | CUB | Avg. |
|---|---|---|---|---|---|
| Zero-shot ($Y$ is attribute vector) [3] (Fig. 3a) | 66.4 | 62.0 | 82.5 | 42.8 | 63.4 |
| Transferring linear offsets in the embedding space | 81.3 | 72.1 | 73.5 | 73.9 | 75.2 |
| Replacing attribute with sample from class (Fig. 3b) | 85.4 | 78.1 | 81.1 | 81.1 | 81.4 |
| $Z$ is sampled from training set, not random (Fig. 3c) | 86.6 | 84.2 | 80.1 | 77.0 | 81.9 |
| Autoencoder instead of denoising-autoencoder (Fig. 3d) | 88.2 | 80.9 | 79.5 | 79.1 | 81.9 |
| Adding $Y$ as input to encoder too (Fig. 2) | 90.5 | 82.5 | 82 | 82.2 | 84.3 |

## 4.4 Are we synthesizing non-trivial samples?

We have observed a significant few-shot performance boost when using the proposed method for sample synthesis when compared to the baseline of using just the few provided examples. But are we learning to generate any significant new information on the class manifold in the feature space? Or are we simply augmenting the provided examples slightly? We chose two ways to approach this question. First, evaluating the performance as more samples are synthesized. Second, visualizing the synthesized samples in the feature space.

Figure 4 presents the accuracy as a function of number of generated samples in both 1-shot and 5-shot scenarios when evaluated on the *mini*ImageNet dataset. As can be seen the performance improves with the number of samples synthesized, converging after $512 - 1024$ samples. For this reason we use $1024$ synthesized samples in all of our experiments. It is interesting to note that at convergence, not only using the 5-shot synthesized samples is significantly better then the baseline of using just the five provided real examples, but also using the samples synthesized from just one real example is better then using five real examples (without synthesis). This may suggest that the proposed synthesis approach does learn something non-trivial surpassing the addition of four real examples.

To visualize the synthesized samples we plot them for the case of 12 classes unseen during training (Figure 5a). The samples were synthesized from a single real example for each class (1-shot mode) and plotted in 2d using t-SNE. As can be seen from the figure, the synthesized samples reveal a non trivial density structure around the seed examples. Moreover, the seed examples are not the centers of the synthesized populations (we verified that same is true before applying t-SNE) as would be expected for naive augmentation by random perturbation. Hence, the classifier learned from the synthesized examples significantly differs from the nearest neighbors baseline classifier that is using the seed examples alone (improving its performance by $20 - 30$ points in tables 1 & 2). In addition, Figure 5b shows visualizations for some of the

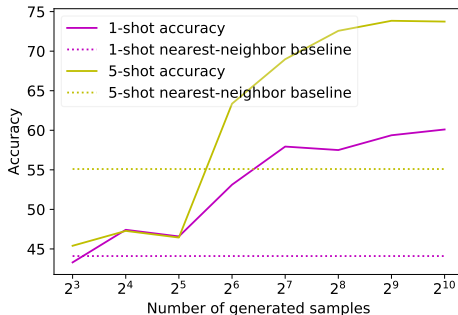

Figure 4: ***mini*ImageNet 5-way Accuracy vs. number of generated samples.** As indicated by the accuracy trend we keep generating new meaningful samples till we reach convergence at $\sim 1K$ samples.

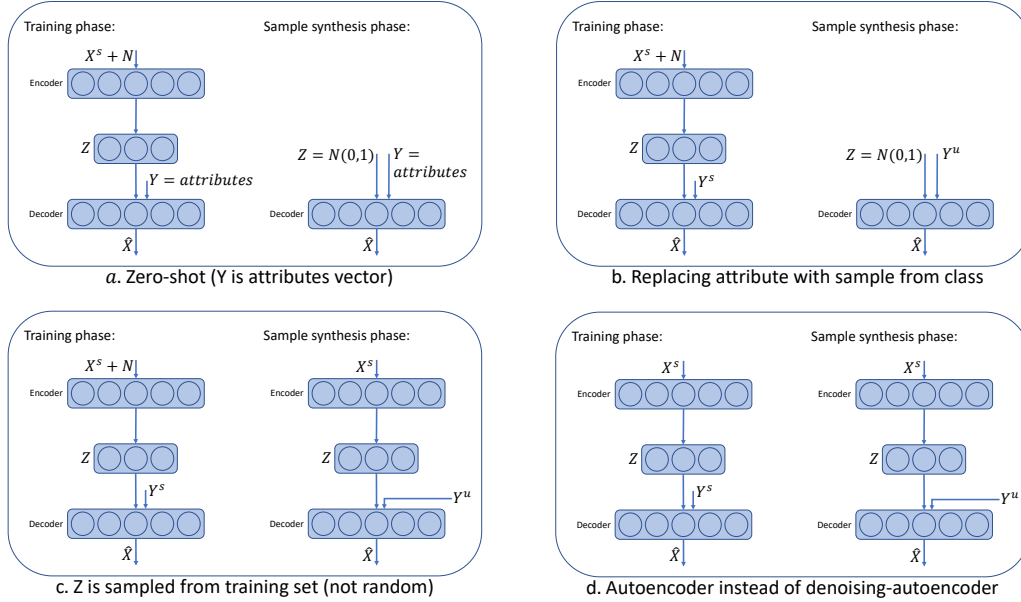

Figure 3: **Different design choices tried.** Classification accuracy for each architecture is presented in Table 4. The final chosen architecture is depicted in Fig. 2.

synthesized feature vectors obtained from a given seed example. The images displayed are the nearest neighbors of the synthesized ones in the feature space.

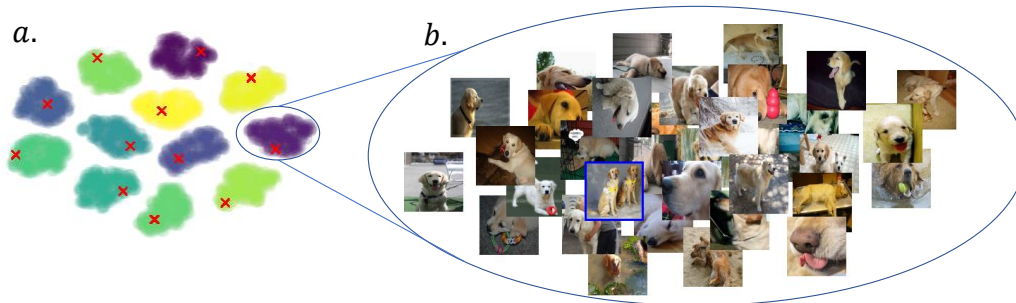

Figure 5: **a. Generated samples for** $12$**-way one-shot.** The red crosses mark the original 12 single-samples. The generated points are colored according to their class. **b. Synthesized samples visualization.** The single image seen at training is framed in blue. All other images represent the synthesized samples visualized using their nearest "real image" neighbors in the feature space. The two-dimensional embedding was produced by t-SNE. Best viewed in color.

## 5   Summary and Future work

In this work, we proposed a novel auto-encoder like architecture, the $\Delta$-encoder. This model learns to generate novel samples from a distribution of a class unseen during training using as little as one example from that class. The $\Delta$-encoder was shown to achieve state-of-the-art results in the task of few-shot classification. We believe that this new tool can be utilized in a variety of more general settings challenged by the scarceness of labeled examples, e.g., in semi-supervised and active learning. In the latter case, new candidate examples for labeling can be selected by first generating new samples using the $\Delta$-encoder, and then picking the data points that are farthest from the generated samples. Additional, more technical, research directions include iterative sampling from the generated distribution by feeding the generated samples as reference examples, and conditioning the sampling of the "deltas" on the anchor example for better controlling the set of transformations suitable for transfer. We leave these interesting directions for future research.

**Acknowledgment:** Part of this research was partially supported by the ERC-StG SPADE grant. Rogerio Feris is partly supported by IARPA via DOI/IBC contract number D17PC00341. Alex Bronstein is supported by ERC StG RAPID. The U.S. Government is authorized to reproduce and distribute reprints for Governmental purposes notwithstanding any copyright annotation thereon. Disclaimer: The views and conclusions contained herein are those of the authors and should not be interpreted as necessarily representing the official policies or endorsements, either expressed or implied, of IARPA, DOI/IBC, or the U.S. Government)

## Footnotes

* The authors have contributed equally to this work

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
