[Reviews · NeurIPS 2018]

Reviewer 1



*** After rebuttal: Thank you for the additional experiment, this satisfies my curiosity and convinces me more of the inability of a linear offset to achieve similar results. It is also encouraging that the reported results compare favourably to the recent paper by Gidaris et al. Finally, the provided explanation fully clarify my understanding of the algorithm, and I feel it's a nice and simple idea. I've raised my score to 7 as a result of all the above. One thing that would be important to clarify is that the feature extractor is trained exclusively on the meta-training set of each dataset considered. Another reviewer misunderstood this fact, and I remember also misunderstanding this during my first pass of the paper. I think the confusion might be due to using the word 'pretrained' which may be interpreted as reusing a feature extractor that was trained on another dataset(s), instead of merely describing a two-stage training phase where first the feature extractor is trained, and then held fixed while the encoder / decoder are trained. It would be useful to reword the relevant parts to avoid misunderstandings. *** Original review: This paper proposes a method for few-shot classification of novel categories that belongs to the family of data augmentation. Specifically, the problem of few-shot classification requires constructing a classifier to distinguish between N new classes, of which only a few examples are available. The data augmentation approach to the problem aims to generate or synthesize additional ‘training’ examples for learning this N-way classifier, directly addressing the data insufficiency which is the cause of the difficulty of this setup. The proposed method, named Delta-encoder, can be used to generate additional examples of a new category given only a single ‘anchor’ example of that category. Intuitively, it works by creating a latent representation of a transformation that can be performed to the anchor example to transform it into something that still plausibly belongs to the same class as the anchor. In other words, it learns what transformations are ‘allowable’ for maintaining the ‘class-yness’ of the anchor example. This idea is the same as in [17], but the authors here propose a different method for achieving it. Notably, while [17] applied linear offsets to the anchor to generate new examples, this proposed method employs a non-linear function for learning these allowable transformations. More specifically, Delta-encoder has an auto-encoder structure where the encoder takes a pair of examples that are known to belong to the same class and is encouraged by its training objective to use the latent space for encoding the ‘deformation’ between those two training examples. Specifically, this latent representation in addition to one of the initial two training examples are used by the decoder to reconstruct the other example. The dimensionality of the latent code is small enough to disallow this structure to function as a standard autoencoder where one of the two examples is merely reconstructed from itself, and the other is ignored. Therefore, what the latent code learns is how one of the two examples needs to be modified in order to be transformed into the other. The hope is that these ‘allowable class-preserving deformations’ that are learned between examples of training classes also are useful class-preserving deformations between instances of new classes. Pros - Well-written, clearly conveys the problem setup and the proposed method - The proposed model is interesting and novel, to the best of my knowledge, and tackles an important problem - They show experimentally that this method performs very well on a wide range of standard datasets. - The section on ablation studies justifies the incremental changes performed on top of a Denoising Autoencoder model (for zero-shot learning) that was the inspiration for the Delta-Encoder. Cons - While [17] and [43] seems to be the most related works, there is no experimental comparison to these methods. I feel this is quite important, and having this additional data point would significantly impact my opinion. - While a “linear offset” version is experimented with in the ablation studies section, I find that the particular linear offset version used there is a particular special case. Specifically, instead of computing the latent code as the vector difference between the anchor point and the input, this could be computed as the output of a linear layer that these two vectors as input (with a trainable weight and bias). I think it would be useful to quantify how well the best possible linear version fairs against Delta-Encoder. - While the paper overall is clearly written, I found lines 126-129 somewhat unclear. In particular, if I understand correctly these lines describe how the Delta-encoder is used at few-shot learning time. In particular, at this time, a single example of each new class is used as the anchor (if k > 1 examples are available of a class, they will in turn be used as the anchor) and the decoder produces additional examples that are hypothesized to belong to this new class, by taking as input this anchor and a Z (the latent code that encodes the deformation to be applied to the anchor). The part that is unclear to me, which is described by these lines, is how this Z is obtained. The quoted lines explain that a number of pairs of examples are encoded into Z’s, where each pair belongs to a training class (not one of the N new classes). This gives a set of { Z_i }’s. The part that is missing is how these are combined to form the Z that will condition the generation of examples of a new class. Is it an average that’s taken? In that case, would it make sense to just take the average over all encoded training pairs (which can be computed during the autoencoder training). - It should be noted that the representation-learning architecture used here is significantly larger than it is in many methods being compared to, casting doubts about the apples-to-apples-ness of these comparisons. Maybe an additional asterisk or note can be included in the table to indicate this (similar to how other aspects are indicated). In particular, Meta-Learner LSTM, MAML, Matching Networks, and Prototypical Networks all use a simple 4-layer CNN architecture. Suggestions - I think that including a description of the overall algorithm in the section that explains the method would really aid the reader in understanding the process. For example something like the following: Given a dataset, the following phases are performed. Phase 1: Train a standard classifier on the set of training classes (often referred to as the meta-training set) of the dataset (all-way classification). These learned weights function as the pre-trained representation used in what follows. Phase 2: Train the encoder and decoder weights. As I understand it, this amounts to randomly sampling pairs of examples that have the same label, and training the autoencoder as described above. Phase 3: Few-shot learning phase. At this point, a ‘test episode’ is presented that contains k examples of each of N new classes. Each of these k examples of a given class will in turn be considered as the ‘anchor’ and the decoder will generate additional images of that class given this anchor and the latent code representing a deformation. Then, given all (real and synthesized) examples, an N-way classifier is trained for this episode. - Regarding Figure 2: A more descriptive and less confusing name for phase 2 (on the right of the figure) might be 'few-shot learning phase' instead of ‘sample synthesis phase’. In particular, as I understand it 'sample synthesis' is the process of decoding, which happens in both phases. I think the key part that distinguishes the two phases is that the former trains the encoder and decoder, while the latter applies the decoder to generate examples of new classes, given only an anchor for these classes. - In related work, according to the proposed distinction between metric learning and meta-learning, Prototypical Networks should be in this meta-learning category too, instead of the metric learning one. Its training objective follows exactly the same structure as Matching Networks, the only difference being the choice of the conditional classifier's form.

Reviewer 2



**Update after rebuttal:** I would like to thank the authors for the responses to my questions. I would like have liked to see more discussion of the details on the P(Z|Y) part they mentioned in the response but I appreciate that this is impossible due to length discussion. I still have reservations about the approach but nevertheless authors have thoroughly investigated alternatives and made a well-reasoned choice on the basis of those considerations. Having read the response and the paper again, and as a vote of confidence in the proposed added discussions about the independence of Z and Y, I have increased my score. The authors propose a new neural network architecture and a corresponding training scheme to solve the problem of one-shot generative modeling. It works by encoding the difference between pairs of images in a bottleneck layer, and then combining this difference information with one of the images it reconstructs the other image. At test time, the architecture can be used to generate further examples of a class from a prototype example.The authors use this model to solve one-shot classification, by first generating additional examples of the given class prototypes, and then training a full classifier on the generated examples. Comment about the overall approach: * My main criticism of this approach is the mismatch of what this architecture is designed to do and how it is ultimately used. * This approach appears to solve a significantly harder problem (one-shot generation) in order to solve an easier problem: one-shot classification. * As a result of this mismatch, the overall method is complicated and it is not and cannot be end-to-end tuned to the task at hand. * Furthermore it is unclear how improving or modifying the training procedure for the delta encoder would effect performance on the classification task. Questions/comments about the Delta-encoder architecture: * In Figure 2 right-hand panel, X_z and Y_s are only needed to create a sample of Z so that it’s from the same distribution as the distribution of Zs during training. Is this correct? Could one regularize the distribution of Z to take a certain shape (say via a KL-regularizer like in beta-VAE) so that the encoder is no longer needed at test time? * [major question] At training time, the two inputs to the decoder Z and Y_s are not statistically independent, yet at test time Z and Y_u are. I’m not sure how significant this is but to me this seems to create a particular mismatch between how we train and how we use the model. Would it be possible to devise an experiment to test how much of a problem this is? Is the distribution of Z class-independent, for example? If we feed the encoder a random X, and then feed the encoder and decoder (A) the same Y from a seen class, vs (B) different Ys from the same seen class, how does the distribution of reconstructed \hat{X} change? In particular, would the top layers of the VGG network still classify these correctly? * What, other than the relatively small size of the bottleneck layer, ensures that the model can’t simply store enough information in Z to restore X without ever using any information about Y at all? Have the authors considered adding an extra regularizer term which simultaneously tries to minimise the information stored in Z about X somehow? Questions about experiments * [major question] The method uses precomputed VGG19 and ResNet features (line 158). Presumably these have been pre-trained on some particular datasets, like ImageNet or Cifar. Presumably this pre-training included classes which are ‘unseen’ classes in the one-shot setting. If this is the case I would say that the method can derive unfair advantage from this as it already knows how to represent the unseen classes. Could the authors comment on this? * In section 3.1 the authors comment on wallclock time but it is unclear to me if these figures include training the classifier once the samples are generated from the decoder? * Otherwise the results seem strong. I am unfamiliar with the accuracy numbers one should expect in this particular application but I have double checked a couple references and the numbers reported here seem to match the numbers published at the referenced papers. In summary, I think this is a well executed and well written paper, I welcome the discussion of other architectures considered and the thorough comparisons with other methods. I am disappointed by the complexity of the solution and particularly the mismatch between what the paper sets out to solve (classification) and the eventual solution (which involves generation). In my view this limits the scope of impact this solution may ultimately have on the field of one-shot classification. This is the main reason I did not give it a higher score. I indicated that my confidence level as 4 as I am unfamiliar with relevant work producing SOTA results in these applications so I could not judge the strength of the reported results with 100% certainty. I also had a few questions about VGG and ResNet pretraining which may alter my assessment. Authors, if limited on space, please focuson on addressing questions marked [major question]. Thanks!

Reviewer 3



------------------After authors' feedback----------------- Authors will update the paper according to my review. After reading other reviews, I still think that it is a good paper. ----------------Before authors' feedback----------------- Summary: I recommend acceptance of this paper, for its quality and clarity. Even if the originality of the algorithm is not outstanding, the results are. Description: $\Delta$-encoder: an effective sample synthesis method for few-shot object recognition addresses few-shot learning through data augmentation. More specifically, the authors designed a neural network that creates synthetic images, in the feature space, by modifying -- or in their words applying deltas to -- unseen real images. The neural network is a particular auto-encoder that works on pairs of images. Finally, the real unseen images and the synthetic ones are used to train a linear classifier. Their approach is evaluated on several datasets and outperforms a wide range of competitive algorithms from the state-of-the-art. In addition, as their approach performs data augmentation in the feature space, it allows to use pretrained neural network to extract the features. Doing so improves significantly the performances and still outperforms others approaches combined with a pretrained feature extractor neural network. The authors also performed an ablation study. Some directions for future works are proposed at the end. Quality: The submission is technically sound. Experimentations are done thoroughly and carefully to be fair against the wide variety of algorithms addressing few-shot or zero-shot learning. A lot of different datasets are used for evaluation. The ablation study reveals interesting findings and justify the design of their algorithm. I like the fact that authors included computation times as it proves their algorithm runs a reasonable time. I found the part analysis the quality of the synthetic samples a bit weak and less convincing than the rest of the paper. In particular, I am not sure that Figure 5a really demonstrates anything. Finally, to be completely fair, the authors should highlight that training the linear classifier with the unseen examples and their associated generated ones is necessary, whereas some other approaches, like the ones using the nearest neighbors, don’t rely on any kind of retraining. In addition, I would have like to see more details on the linear classifier and its implementation and a comparison to kNN to see how much replacing the classic 1-NN by a linear classifier improved the result. Overall, the quality is good, but additional details on the linear classifier should be given in the final version. Clarity: The paper is very well written, pleasant to read and well organized. The numerous illustrations boost the clarity of the paper. Results are clearly stated and are easy to review. The implementation is very well detailed. However, no details is given on the linear classifier (cost function?). This prevents to exactly reproduce the results. On the other hand, authors will share the code with the final version. To summarize, clarity is outstanding. Originality: Although simple and quite standard, the proposed algorithm achieves very good results. Related work section is well organized and reviews all the different approaches to few/zero-shot learning. The authors justified their approach using results in the previous works. As they mentioned, they incrementally built on top of [3]. Overall, originality is ok. Significance: Few-shot learning is a very important field in machine learning that has drawn a lot of attention recently. As the proposed algorithm outperforms by a large margin the other algorithms, it constitutes a good advance in the state of the art for few-shot learning. Line 241: then -> than